# Biomechanical Behaviour and Biocompatibility of Ureidopyrimidinone-Polycarbonate Electrospun and Polypropylene Meshes in a Hernia Repair in Rabbits

**DOI:** 10.3390/ma12071174

**Published:** 2019-04-10

**Authors:** Marina Gabriela M. C. Mori da Cunha, Lucie Hympanova, Rita Rynkevic, Tristan Mes, Anton W. Bosman, Jan Deprest

**Affiliations:** 1Centre for Surgical Technologies, Group Biomedical Sciences, KU Leuven, 3000 Leuven, Belgium; biamori@gmail.com (M.G.M.C.M.d.C.); lucie.hympanova@upmd.eu (L.H.); r.rynkevic@gmail.com (R.R.); 2Department of Development and Regeneration, Woman and Child, Group Biomedical Sciences, KU Leuven, 3000 Leuven, Belgium; 3Institute for the Care of Mother and Child, Third Faculty of Medicine, Charles University, 14700 Prague, Czech Republic; 4INEGI, Faculdade de Engenharia da Universidade do Porto, Universidade do Porto, 4099-002 Porto, Portugal; 5SupraPolix BV, 5611 Eindhoven, The Netherlands; mes@suprapolix.com (T.M.); bosman@suprapolix.com (A.W.B.); 6Pelvic Floor Unit, University Hospitals KU Leuven, 3000 Leuven, Belgium

**Keywords:** mechanical load, stress shielding, absorbable implants, mesh integration, muscle atrophy, fatty infiltration

## Abstract

Although mesh use has significantly improved the outcomes of hernia and pelvic organ prolapse repair, long-term recurrence rates remain unacceptably high. We aim to determine the in vivo degradation and functional outcome of reconstructed abdominal wall defects, using slowly degradable electrospun ureidopyrimidinone moieties incorporated into a polycarbonate backbone (UPy-PC) implant compared to an ultra-lightweight polypropylene (PP) textile mesh with high pore stability. Twenty four New-Zealand rabbits were implanted with UPy-PC or PP to either reinforce a primary fascial defect repair or to cover (referred to as gap bridging) a full-thickness abdominal wall defect. Explants were harvested at 30, 90 and 180 days. The primary outcome measure was uniaxial tensiometry. Secondary outcomes were the recurrence of herniation, morphometry for musculofascial tissue characteristics, inflammatory response and neovascularization. PP explants compromised physiological abdominal wall compliance from 90 days onwards and UPy-PC from 180 days. UPy-PC meshes induced a more vigorous inflammatory response than PP at all time points. We observed progressively more signs of muscle atrophy and intramuscular fatty infiltration in the entire explant area for both mesh types. UPy-PC implants are replaced by a connective tissue stiff enough to prevent abdominal wall herniation in two-thirds of the gap-bridged full-thickness abdominal wall defects. However, in one-third there was sub-clinical herniation. The novel electrospun material did slightly better than the textile PP yet outcomes were still suboptimal. Further research should investigate what drives muscular atrophy, and whether novel polymers would eventually generate a physiological neotissue and can prevent failure and/or avoid collateral damage.

## 1. Introduction

Meshes are used in several conditions, including hernia, and pelvic organ prolapse (POP), where they have been shown to significantly improve outcome as compared to native tissue repair. [1,2]. However, even after the use of implants, long-term recurrence rates are close to 30%, which is unacceptably high [3,4].

The most frequently used meshes are textile in nature and typically made from non-resorbable polypropylene (PP) [5]. Meshes may induce implant-related complications (IRCs) such as pain, infection, erosion, rejection of the implant and adhesion formation [6]. Several factors have been associated with IRCs, including the textile properties or polymer choice, the mechanical behaviour of the mesh, the nature of the host immune response, the surgical experience, and other patient factors [7,8,9]. Over time, there has been a trend for reducing the amount of foreign material, by increasing the pore size and reducing the filament diameter [6,8,9]. Despite the use of lighter meshes, IRCs have not been completely irradiated yet [6]. Essentially, durable implants inherently induce a chronic inflammatory reaction [8,10], whereas they do not necessarily induce functional replacement tissue [6,8]. Typically, they are stiffer compared to native tissue, and may compromise normal function [11]. In addition to local wound healing problems, non-compliant implants may also induce stress shielding. This occurs when two objects are physically connected and the stiffer material (in this case mesh) bears the majority of the load. The less stiff material (in this context the soft tissue) is shielded from the load, and hence undergoes maladaptive remodelling characterized by degeneration and atrophy [12]. Stress shielding may increase the rate of mesh exposure following vaginal implantation [12]. In abdominal wall surgery, it remains unclear how much the mismatch between biomaterial and host tissue contribute to its failure [13,14]. However, the loss of mechanical load signalling was reported to impair fibroblast biology in rats, and the resultant collagen abnormality was found to be the cause for the recurrence [15]. The use of biological meshes was expected to overcome some of these issues, due to the ability of these implants to be remodelled and revascularized. However, their inconsistent properties [16] make it difficult to predict their degradation profile [17].

Because of all the aforementioned problems, the development of novel materials is being encouraged by international authorities [18]. As a first strategy, the use of absorbable polymers is currently being (re)considered. When using these, it is logical to choose absorbable materials that degrade with appropriate kinetics to match the healing process [17,19]. Degradation that is too rapid may lead to recurrence [17,20] and overly slow degradation may still induce a prolonged host response. Supramolecular polymers are materials that have tunability and reversible non-covalent interactions, which enable modular and generalizable platforms with tunable mechanical, chemical and biological properties [21]. Mapping the degradation in vivo of promising supramolecular materials paves the way for screening and selecting materials for various clinical implantation sites. A second modification may be to replace textile fabrics by electrospun ones, which mimics the extracellular matrix-like network. This promotes cellular infiltration, supports cell adhesion and stimulates the production of a natural extracellular matrix [22,23]. Third, one could choose materials that have a pre-implantation compliance within the range of native tissues. Combining a degradable material within a compliant electrospun implant might lead to better integration into the host. 

In a previous study, we have tested the compliance of a hydrogen bonded supramolecular polymer comprising ureidopyrimidinone (UPy) moieties incorporated into a polycaprolactone (PCL) backbone in rats and rabbits. Though UPy-PCL explants were as compliant as native tissue [24], repairs failed in half of the animals early on, due to rapid degradation of the implant [20]. Herein, we replaced the PCL backbone with an aliphatic polycarbonate (PC) backbone since the degradation of PC takes longer than polyesters [25,26]. Moreover, they have elastomeric characteristics while remaining biocompatible [27]. It has been shown in adult sheep that in vivo degradation of a PC-based UPy-polymer in an electrospun pulmonary valve conduit is slower than that of a PCL-based one and it can last at least six months [28]. Therefore, we tested these novel electrospun UPy-PC implants and determined the in vivo degradation and functional outcome, when used for abdominal wall reconstruction following induction of a full-thickness defect that was either bridged by an implant (referred to as “gap bridging”), or primarily sutured (native tissue repair) and reinforced by an implant. 

## 2. Results

### 2.1. Dry Mesh

The stiffness and Young’s modulus of dry mesh and native tissue are displayed in Table 1. PP meshes were significantly stiffer than UPy-PC mesh. Conversely, the stiffness and the Young’s modulus of UPy-PC mesh was in the range of that of native tissue.

### 2.2. Clinical Findings and Gross Anatomy

No clinical complications were observed, including the absence of obvious hernia recurrence. Visually, UPy-PC implants were partly degraded from 90 days onwards. However, there was more degradation in the gap bridging than in the reinforcement sites, and more so at 180 than 90 days. In the gap-bridging defects, the PP-implants were 15–20% smaller, with no difference between time points, see Table 2. UPy-PC implants barely changed from baseline either over time (−5% to +4%; Table 1). Whereas, on initial inspection, no failure of the implant was suspected; after skin removal in 2/6 (33.3%) of the UPy-PC gap-bridging sites there was limited bulging and a thinner appearance in the implant area. We categorized them as subclinical hernias. This coincided with the absence of a visually persistent implant; however, with replacement by a thin layer of connective tissue. In reinforcement sites, bulging did not happen, although there was also degradation on inspection.

### 2.3. Biomechanical Testing of the Explants

Young’s moduli of the explants are displayed in Figure 1. Similar trends over time were observed in both models. Initially (d30) UPy-PC and PP explants presented physiological compliance. However, PP became stiffer than native tissue at 90 days and by 180 days both were stiffer. The UPy-PC explants where herniation was present showed compliance comparable to those without.

### 2.4. Inflammatory Response and Neovascularization

Implant types and the presence of the polymer could be easily recognized upon hematoxylin and eosin (H&E) staining. The PP filaments were surrounded by mild inflammation, see Figure 2A. UPy-PC nanofibres were visible at ×20 magnification, all surrounded by inflammatory cells. From 90 days onwards, those areas got scarcer and thinner. At 180 days, the implant material was disconnected as islands of remnant material, again locally surrounded by foreign body giant cells (FBGC). Although this is a more a semi-qualitative impression, there seemed to be less UPy-PC material in the gap bridging than in the reinforcement sites at 90 and 180 days. Based on the quantification of the FBGC, the UPy-PC group induced a more vigorous foreign body reaction than PP, irrespective of the surgical defect type, see Figure 2B. We further investigated the number of macrophages around the implants. In the UPy-PC group, macrophages were deep in between the microfibres at all time points, see Figure 3. In the PP group, macrophages were observed only around the filaments. There were significantly more macrophages in the UPy-PC explants at 30 days in both models and at 180 days in the reinforcement model.

Vascularization was investigated by expression of CD34, see Figure 4. The number of vessels was similar around both implants at all time points, except at 180 days in the gap-bridging model. At this late time point, we observed significantly less vascularization in the PP explant compared to UPy-PC explants.

### 2.5. Musculofascial Characteristics of the Explants

We observed muscle atrophy with intramuscular fatty infiltration in both models and for both materials, see Figure 5. The amount of connective tissue in the abdominal wall tended to be lower in the PP than in the UPy-PC animals, yet this was only significant at 180 days in the gap-bridging model and at 30 and 90 days in the reinforcement model. At 180 days, there was a significantly lower muscle volume for both materials and both models, compared to native tissue. In the UPy-PC explants from animals with subclinical herniation, sparse polymer elements were present, yet they mostly consisted of a thin layer of connective tissue, see Appendix A.

## 3. Discussion

We studied the short- and long-term morphologic, integration and biomechanical effects of abdominal wall reconstructive surgery with two completely different implants, i.e., electrospun absorbable UPy-PC with initial physiologic compliance and a non-resorbable textile PP, which is ultra-lightweight with stable pores, yet stiffer than native tissue. We challenged these in two surgical models, one for abdominal wall reconstruction (gap bridging) and one for reinforcement of a primarily sutured abdominal wall defect. The most important findings are: (1) UPy-PC is degraded after 180 days, and failed at 180 days in one out of three of the gap-bridged defects, in contrast to PP which persists and did not fail in a single case; (2) both implants compromised the compliance of the abdominal wall at 180 days; (3) coinciding with induced significant muscle atrophy; (4) PP explants displayed a significantly lower amount of adjacent fibrocollagenous neotissue compared to UPy-PC at 30 and 90 days in the reinforcement model and at 180 days in the gap-bridging model; (5) UPY-PC explants showed better integration than PP.

Bioresorbable scaffolds should provide mechanical strength until sufficient mature neo-tissue is formed that can bear the full mechanical force of the abdominal wall and promote a fast and organized host response with minimal inflammation [17]. Ideally, that tissue should have a physiological compliance [13,29,30], which, in turn, enhances the adhesion and proliferation of myoblasts, prevents stress shielding [31] and altogether, leads to greater patient comfort [13,32]. We have previously reported that UPy-PCL explants have biomechanical properties comparable to that of native tissue when used as a reinforcement of a primarily sutured defect [24]. However, when used to bridge a large gap, the failure rate with PCL polymer was as high as 50% at 30 days after implantation [20]. Therefore, we explored herein, a comparable construct but based on another polymer. We used the supramolecular polycarbonate polymer which is slowly degradable and that can have physiological compliance due to its elastomeric characteristics [27]. Apparently, this strategy could not address hernia recurrence nor lead to a physiological compliant neotissue in the long term. Moreover, UPy-PC explants exhibited more new collagen deposition and sustained neovascularization compared with PP meshes, which are important steps for the integration of the mesh into the abdominal wall [33].

Although the herniation rate was lower than when polycaprolactone was used, failure in one out of three subjects at 180 days is clinically still unacceptable. Failure coincided with obvious degradation and neo-tissue formation that was apparently insufficient to bear the mechanical load of the abdominal wall. Degradation of the implants was faster in gap-bridged defects than in reinforced areas. A higher load may have changed the conformational strain energy and morphology of the implant, affecting the stability of the polymer [34]. In addition, it may lead to cracking of the microfibres, hence an increase in the surface area for the oxidative degradation induced by the host inflammatory response. Theoretically, to solve this drawback, one could modify the degradation process by using a slower degrading UPy elastomer and take advantage of its potential for bioactivation and, for example, adding factors to accelerate tissue ingrowth [23,35,36]. The degradation process is also modulated by the vigour and nature of the inflammatory response. In contrast to conventional aliphatic polycarbonates, UPy-PC is not degraded via aqueous or enzymatic hydrolysis, but via an oxidative pathway [37]. Oxidative degradation is mediated by reactive oxygen species that are secreted by macrophages, neutrophils and FBGC that are in contact with the implant [38]. As long as UPy-PC remained visible in the explant, the above cell types were abundantly present; therefore, logically, they may have contributed to the degradation process. The relatively low numbers of inflammatory cells around the permanent PP fibres may be due to the ultra-lightweight nature, the large pore size and the stability of the textile construct, which have been shown to induce a limited vigorous inflammatory response [39].

The most striking observation, according to us, is the progressive decrease of muscle tissue. This is similar to what is seen in mechanically-unloaded muscles leading to disuse atrophy and muscle fiber changes [40]. One mechanism to explain this maladaptive remodelling response may be neotissue stiffness leading to disuse and stress shielding. It is well known that mechanical loading of skeletal muscle post-injury triggers many beneficial responses that enhance skeletal muscle regeneration and reduce fibrosis [41]. Under disuse, fibro-adipogenic precursors proliferate, differentiate into adipocytes and extracellular matrix-producing fibroblasts; thereby resulting in the accumulation of intramuscular fat and fibrotic tissue [42,43]. Moreover, the accumulation of fat tissue in the muscle inhibits its regenerative potential because it suppresses the proliferative capacity of satellite cells and may also inhibit the activity of macrophages [44]. Stress shielding leads to thinning and degradation of the involved tissues due to a breakdown of key structural proteins like collagen and elastin [45]. The loss of muscle fibres and replacement with fatty and fibrous tissue leads to muscle weakness [46] and this process is very difficult to reverse [47]. Another potential reason for this maladaptive response is chronic inflammation [48,49]; however, we did not observe a strong inflammatory response in the PP group, which was equally affected by muscle atrophy. The longer-term effect of muscle atrophy remains to be studied. We speculate that atrophy may, over time, result in mechanical failure or recurrent incisional hernia formation later on, since fascial pathology [50], muscle atrophy and degeneration have all been implicated in hernia pathophysiology [51]. Moreover, it has been suggested that defective collagen synthesis may play an important role in hernia recurrence [9] since one-third of recurrences occur after three years of mesh implantation for incisional hernia repair [52].

There are some limitations associated with the current study that deserve consideration. First, it does use an acute incisional repair model, which does not recreate the chronic pathology and wound environment of clinical herniation. Second, we compared two constructs that were different in many aspects, such as their durability, their textile or electropun nature, as well as their biomechanical properties. Therefore, the experiments do not allow for a mechanistic conclusion about the nature of our observations. We did not control for variables determining the implant properties, such as polymer load (e.g., a thicker mesh or fibres), format of the implant, which would have allowed to test factors that may explain the observations made. In addition, we should have compared outcomes of aliphatic PC with and without UPy motifs. Lastly, this type of surgical experiment does not permit a longitudinal study. On the other hand, to our knowledge this is the first experiment documenting the morphology and compliance of the abdominal wall in the long term, when being reconstructed with either ultra-lightweight polypropylene and novel experimental UPy-PC implants. In addition, it shows that novel compelling strategies for implant design do not necessarily solve old problems.

## 4. Material and Methods

### 4.1. Meshes

Ureidopyrimidinone-polycarbonate polymer was obtained from SupraPolix BV (Eindhoven, the Netherlands) and electrospun by Coloplast A/S (Humlebaek, Denmark). All implants were from the same production lot. Details regarding polymers spinning and sterilisation were described previously [20]. The characteristics of the meshes used in this study are described in Table 1. UPy-PC and PP meshes (n = 6/group) were cut into 50 mm × 10 mm strips along the longitudinal axis of the sterile implant, with respect to the pore pattern, and underwent uniaxial testing for determination of comfort zone stiffness and Young’s modulus, see Appendix A.

### 4.2. Study Design and Surgical Procedure

Eighteen adult New Zealand rabbits (3500 g) were randomly divided into two groups, according to the type of mesh (UPy-PC or PP) used. In each rabbit, four lesions were induced in the abdominal wall: two incisional hernia-type full-thickness defects that were primarily sutured and two full-thickness 4.5-cm² defects, see Figure 6. All were covered by the mesh of interest (40 mm × 25 mm). These defects and repairs are meant to mimic two surgical conditions, a primary fascial defect repair reinforced with mesh (referred to as “reinforcement”) and one where a full-thickness abdominal wall defect is substituted locally by mesh (referred to as “gap bridging”). Rabbits were harvested at 30, 90 and 180 days for studying the integration process of the mesh into the host (n = 3 does per time point; 4 surgical sites and explants per animal, two of each surgical defect type). This experiment hence reports on 6 explants per mesh type/model and per time point overall.

All animals were treated in accordance with current national guidelines on animal welfare and our experiment was approved by the Ethics Committee for Animal Experimentation of the KU Leuven (Project number P064/2013). Anaesthesia and analgesia procedures are described in detail in Appendix A. The surgery was previously described in detail [53].

### 4.3. Clinical Examination and Sizing of the Implants

Rabbits were clinically examined daily for the first week and twice monthly later on. Gross anatomical findings were evaluated during harvesting and classified as hernia recurrence, fluid collection, infection and erosion (i.e., loss of epithelial integrity).

Pictures of the surgical sites were taken with a sterile ruler alongside at D0 and at the day of explantation to allow calculation of dimensional changes. Measurement of mesh dimensions was done later offline, using the ruler to calibrate and image J software to trace the outlines. Contraction (i.e., shrinkage) was defined by the ratio of the implant at the day of the surgery and the day of explantation expressed as a per cent of the initial dimensions (implant at day 0/implant at the day of explantation × 100%). Contraction of the mesh was only calculated for implants from animals without herniation.

The explant was cut into two strips of at least 1-cm width, longitudinal to the long axis of the animal: One for tensiometry and one for histology. A sample of the abdominal wall away from the implant was harvested as a control for histology and biomechanical analysis.

### 4.4. Biomechanical Testing of Explants

Explants were cut into 30 mm × 10 mm strips along the longitudinal axis of the implant, with respect to the pore pattern, and for explants, along the body axis. A total of six specimens were obtained for each explant at each time point. Biomechanical testing was performed on a uniaxial Zwick tensiometer with a 200-N cell load set up (Zwick GmbH & Co. KG, Ulm, Germany). Specimens were clamped tension-free with 0.5 cm within the clamps. Zero elongation was defined as the clamp-to-clamp distance at preload (0.1 N). Cross-sectional dimensions were measured at three sections through the explant using an analogue caliper. Afterwards, the specimen was elongated with a speed of 10 mm/min up to failure. The test was ended when the load fell below 60% of the maximum load. The strain was calculated by dividing the elongation by the cross-sectional area. The Young’s modulus in the comfort zone was defined as the minimum modulus noted over an interval of 10–20% of elongation. Clinically, the comfort zone of the stress–strain curve is considered to be within the physiological range of deformation. Contralateral muscle specimens were used as a control.

### 4.5. Histology

Formalin-fixed specimens were embedded in paraffin and cut into 5-μm thick slices. Quantification of foreign body giant cells (FBGCs) was performed on hematoxylin and eosin (H&E) stained sections. The amount of muscle and fascial tissue of the explant was assessed semi-quantitatively on Masson’s Trichrome stained sections. The latter stains connective tissue blue and muscle red. Immunohistochemistry was performed to detect neovascularization (CD34, Abcam, Cambridge, UK) and macrophages (RAM-11; MO63; Dako Corp., Carpinteria, CA, USA). For further information, please check the Appendix A.

All morphometric analyses were performed using ImageJ software. Ten non-overlapping fields along the complete interface area of interest were assessed per staining. The area between fascial layers of the abdominal wall was defined as the region of interest for Masson´s Trichrome (×25 magnification) and the interface between the implant and host tissue for H&E, CD34 and Ram-11(200× magnification). For quantification of FBGCs, cells were manually quantified using a cell counter plugin. The digital colour images were segmented (colour deconvolution plugin) and further binarised in order to measure the percentage of the area stained in blue and red (connective tissue and muscle—Masson’s trichrome) or brown (immunohistochemistry). Semi-quantitative readings were done by two observers blinded to the treatment group, though that was, in retrospect, impossible since they appear completely different.

### 4.6. Statistics

All analyses were done with Graphpad Prism 8.0. For FBGC, RAM 11 and CD34 analyses, comparison between groups at different time points were performed using a t-test when data was parametric or Mann-Whitney for non-parametric data. Other readouts, which also included the native tissue group, were analyzed by one-way ANOVA followed by a Tukey test. Data are reported as individual values or mean ± standard deviation (SD). Statistical significance level was defined as p < 0.05.

## 5. Conclusions

In this study, we demonstrated that abdominal wall reconstruction with either electrospun or textile PP implants was associated with muscular atrophy and fatty infiltration in the area of implantation. Eventually, both implant materials induce a response that generates tissues that are less compliant than native tissues, though the time course and magnitude of the response is dependent on the material used. The novel material did partly better than the textile polypropylene; however, yet outcomes were still suboptimal. Further research should investigate what drives muscular atrophy, and whether modifications by bioactivation and/or novel polymers would eventually generate a physiological neotissue, prevent failure and/or avoid collateral damage under the form of muscular degeneration.

## Figures and Tables

**Figure 1 materials-12-01174-f001:**
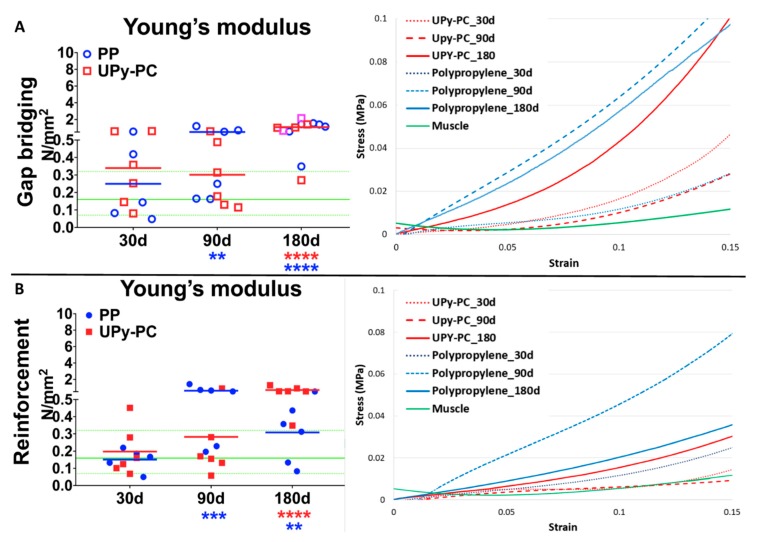
Biomechanical analysis. The left panels display the individual observations of Young’s modulus in the comfort zone of the abdominal wall explants of rabbit after implantation using polypropylene (PP, blue) or UPy-PC (red) in a gap bridging (**A**) and reinforcement (**B**) hernia model. The horizontal green full and dotted line represents the mean, the minimum, and the maximum values of native muscle tissue (NM). The right panels display the curves of stress–strain in the range of 0–20% of elongation. Gap briding: PP was stiffer than native muscle tissue (NM) at 90 and 180 days and UPy-PC was stiffer than NM at 180 days. The compliance of UPy-PC explants where herniation was present fell in the same range as the ones without. Reinforcement: PP showed a significantly stiffer abdominal wall than that of NM at 90 and 180 days and UPy-PC was significantly stiffer than NM at 180 days. Blue (PP) or red (UPy-PC) asterisks represent the difference compared to native tissue. ** *p* ≤ 0.01, *** *p* ≤ 0.001 and **** *p* ≤ 0.0001. Herniated samples are marked by a lighter colour (pink). PP: Polypropylene; UPy-PC: electrospun ureidopyrimidinone polycarbonate.

**Figure 2 materials-12-01174-f002:**
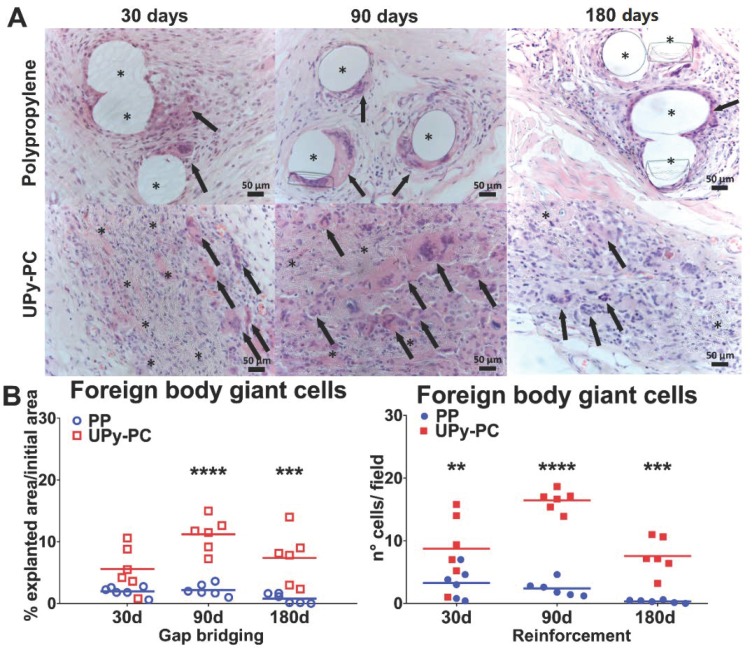
Foreign body response: (**A**) Representative images of H&E stain (200× magnification) at the mesh–tissue interface. Mesh fibres, large white spaces in the case of polypropylene and thin fibres in the case of UPy-PC, are marked by asterisks. Foreign body giant cells (FBGC; black arrows), were more abundant in UPy-PC specimens. In the areas with a lower presence of UPy-PC fibres, due to degradation, FBGC were also less present. (**B**) Graphs of FBGC counts of gap bridging (left) and reinforcement (right). At all the time points and in both models, counts of FBGC were higher in UPy-PC explants (red). ** *p* ≤ 0.01, *** *p* ≤ 0.001 and **** *p* ≤ 0.0001. FBGC counts from UPy-PC recurrence are represented by light pink. PP: Polypropylene; UPy-PC: electrospun ureidopyrimidinone polycarbonate.

**Figure 3 materials-12-01174-f003:**
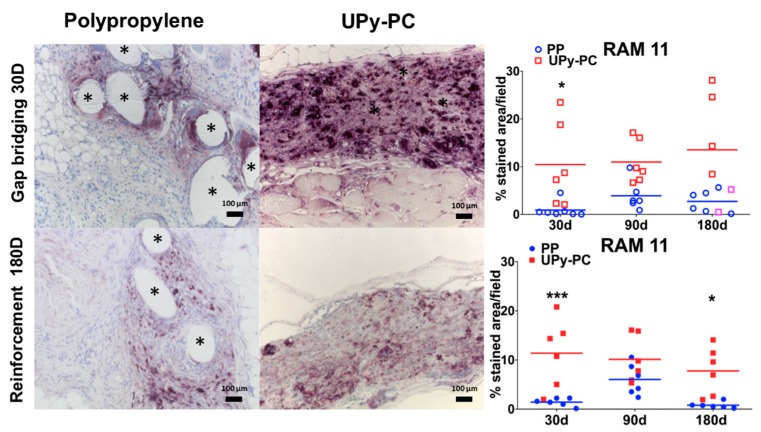
Inflammatory response. Representative images of RAM11 immunostaining at 200× magnification (macrophages stained brown). Mesh filaments are marked by asterisk. A higher abundance of macrophages was around UPy-PC filaments at 30 days in both models, and at 180 days in the reinforcement model. * *p* ≤ 0.05, *** *p* ≤ 0.001 Herniated samples are marked by a lighter colour (pink). PP: Polypropylene; UPy-PC: electrospun ureidopyrimidinone polycarbonate.

**Figure 4 materials-12-01174-f004:**
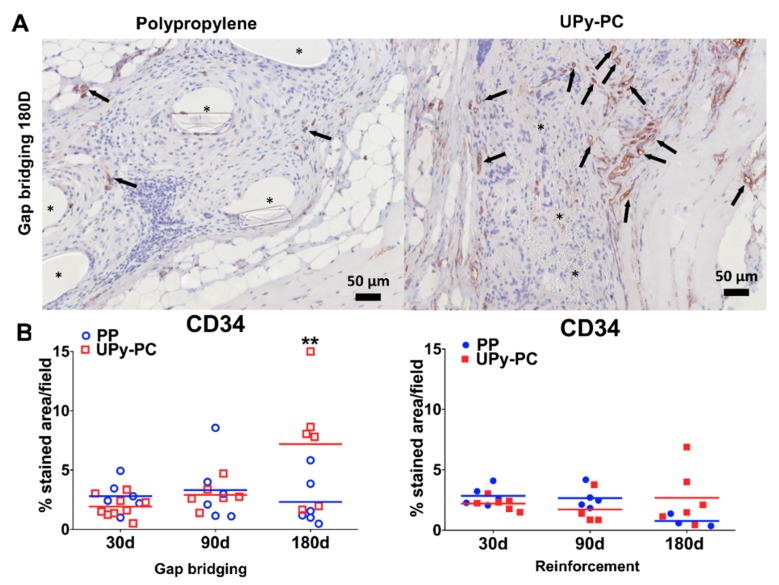
**(A**) Representative images of CD 34 immunostaining at 200× magnification (endothelial cells stained brown). (**B**) Graphs of CD34 positive counts of gap bridging (left) and reinforcement (right). More abundant neovascularization was visible in UPy-PC at 180 days in the gap-bridging model. No significant difference was observed in the reinforcement model. Herniated samples are marked by a lighter colour (pink). ** *p* ≤ 0.01. PP: Polypropylene; UPy-PC: electrospun ureidopyrimidinone polycarbonate.

**Figure 5 materials-12-01174-f005:**
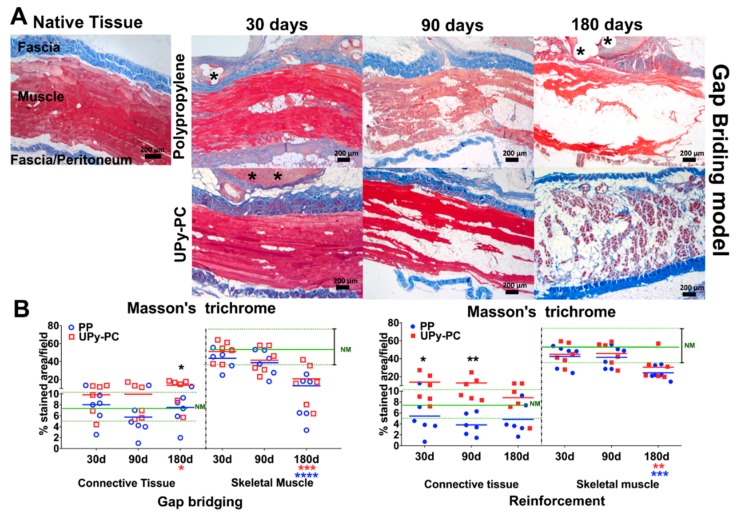
Musculofacial content: (**A**) Representative images of Masson´s Trichrome staining (50× magnification) of the abdominal wall of rabbits after implantation of polypropylene and electrospun UPy-PC at 30, 90 and 180 days in the gap-bridging model. Connective tissue stained in blue and muscle in red. Mild signs of muscle atrophy and intramuscular fatty infiltration starts at 30 days and become more intense at 90 and 180 days in both groups. (**B**) Polypropylene explants showed a significant loss of connective tissue compared to UPy-PC at 180 days in the gap-bridging model (left) and at 30 and 90 days in the reinforcement model (right). The horizontal green full and dotted line represents the mean, the minimum and the maximum stiffness of native muscle tissue (NM). * *p* ≤ 0.05, ** *p* ≤ 0.01, *** *p* ≤ 0.001, and **** *p* ≤ 0.0001. Herniated samples are marked by a lighter colour (pink). PP: Polypropylene; UPy-PC: electrospun ureidopyrimidinone polycarbonate.

**Figure 6 materials-12-01174-f006:**
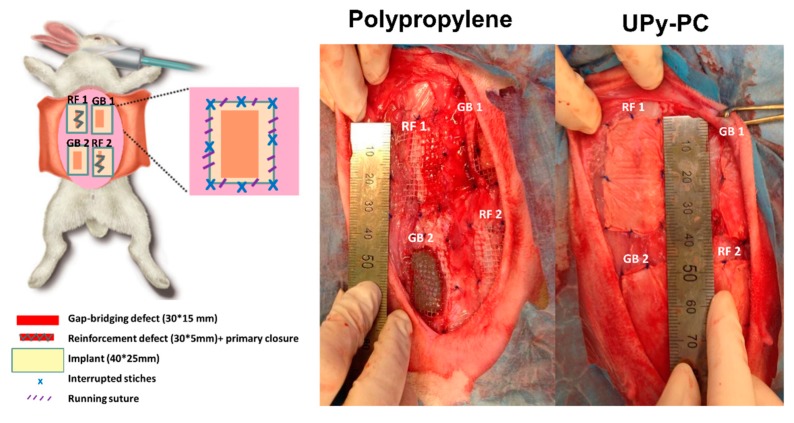
On the left, a scheme of the surgical procedure in a rabbit. After raising skin flaps, two longitudinal 3.0-cm-long 5-mm-wide full-thickness defects were made in the upper right and lower left quadrants of the lateral anterior abdominal wall and parallel to the midline. These incisions were primarily closed by a continuous 4/0 polypropylene (Prolene, Ethicon, Dilbeek, Belgium) suture at an interrun distance of 0.5 cm in the reinforcement group. In the upper left and lower right quadrant, a 30 mm × 15 mm full-thickness defect was induced and overlaid tension free by the implant of interest (gap bridging). The mesh was fixed at the four corners and half way along each side with PP sutures, and further fixed with a running poliglecaprone 25 (Monocryl, Ethicon) 4-0 around the implant at an interrun distance of 5 mm. Meshes were 25 mm × 40 mm, hence oversizing the defects. On the right, a representative image of the surgical site immediately after implantation of polypropylene and UPy-PC mesh. RF1 and 2: reinforcement sites; GB1 and 2: gap bridging sites. UPy-PC: electrospun ureidopyrimidinone polycarbonate.

**Table 1 materials-12-01174-t001:** Characteristics of meshes used in the study. Biomechanical parameters are displayed as mean ± SD.

Parameter	Native Muscle	Macroporous Polypropylene (Restorelle^®^)	Electrospun Polycarbonate Ureidopyrimidinone (UPy-PC)
Mesh thickness (µm)	-	300	250–300
Fiber size (µm)	-	80.0	1.86–3.2
Density (g/m^2^)	-	19.0	41
Comfort zone stiffness (N/mm)	0.20 ± 0.14 ^#^	0.57 ± 0.06 ^#^ ***	0.07 ± 0.06 ***
Comfort zone young’s modulus (N/mm^2^)	0.17 ± 0.09 ^#^	3.94 ± 0.3 ^#^ ****	0.38 ± 0.015 ****

^#^ indicates a significant difference compared to native tissue (*p* < 0.001); *** and **** indicate a significant difference between the treatment groups (PP vs UPy-PC), (*p* < 0.0001) and (*p* < 0.00001) respectively. PP: Polypropylene; UPy-PC: electrospun ureidopyrimidinone polycarbonate.

**Table 2 materials-12-01174-t002:** Gross anatomy of gap bridging and reinforcement surgical sites in the polypropylene (PP) and Ureidopyrimidinone-polycarbonate (UPy-PC) group. Percentages of contraction are displayed as mean ± SD.

Time Point		30 days		90 days		180 days	
Group		PP	UPy-PC	PP	UPy-PC	PP	UPy-PC
**Gap bridging**	Signs of degradation	0% (0/6)	0% (0/6)	0% (0/6)	50% (3/6)	0% (0/6)	100% (6/6)
	Recurrence of hernia	0% (0/6)	0% (0/6)	0% (0/6)	0% (0/6)	0% (0/6)	33% (2/6)
	Contraction (-)	−20.8 ± 6.43%	−5.4 ± 4.7%	−15 ± 5.2%	+4 ± 10.4%	−15 ± 7.6%	+1.7 ± 13.5%
**Reinforcement**	Signs of degradation	0% (0/6)	0% (0/6)	0% (0/6)	18.6% (1/6)	0% (0/6)	100% (6/6)
	Recurrence of hernia	0% (0/6)	0% (0/6)	0% (0/6)	0% (0/6)	0% (0/6)	0% (0/6)
	Contraction (-)	+12.7 ± 5.9%	+12.7 ± 6.2%	+7.6 ± 5.3%	+21.3 ± 10.1%	+1.52 ± 7.8%	+22.1 ± 9.8%

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
