# Peer review of "Biomechanical Behaviour and Biocompatibility of Ureidopyrimidinone-Polycarbonate Electrospun and Polypropylene Meshes in a Hernia Repair in Rabbits"

_materials, 2019, doi:10.3390/ma12071174_

Reviewer 1 Report

In this manuscript, a novel absorbable prosthesis for abdominal hernia is developed, based on electrospun ureidopyrimidinone-polycarbonate (UPy-PC), and compared to a lightweight polypropylene mesh. UPy-PC and PP mesh explants are studied in terms of biomechanical properties. Moreover, muscle atrophy and fat infiltration in host tissue are evaluated. The manuscript presents an interesting idea and valuable results are obtained. However, I have some concerns which the authors should address:

 - Why did you choose Restorelle® mesh as a comparison? Restorelle® is generally used for pelvic organ prolapse repair and not for hernia surgery. The aim of the work seems to be to reduce surgical recurrence and post-surgery pain and discomfort in case of abdominal hernia patients. In this sense, the comparison of UPy-PC with a well-consolidated PP mesh for hernia repair would have been more appropriate. Could you discuss in the Introduction which are the similarities/differences between Restorelle® and other PP hernia meshes? (See, for example: J. Biomed. Mater. Res. Part B: Appl. Biomater., 105B: 892-903, 2017).

On the other hand, the authors are considering the use of absorbable polymers for hernia repair. In this respect, the comparison of degradation behavior in vivo should have been carried out also with a biological mesh. Please, comment on this point and mention the role of biological meshes in the Introduction/discussion too.

 - Results: Biomechanical testing. Please, add stress vs. strain plots for PP and UPy-PC explants in the range 0-20% of elongation. Stiffness and Young’s modulus are not enough to compare mesh mechanical behavior.

 - Section 4.4 Biomechanical testing.

1. Why did you test mesh explants in only one direction? Do you expect both PP and UPy-PC to be mechanically isotropic? Did you get any data on PP and UPy-PC meshes before implant?

2. Can you explain better how comfort zone stiffness and Young’s modulus were determined? Please, note that I was not able to find any Supplement in my reviewer center.

 - Section 4.6 Statistics. This section is too concise and it is not clear which groups were taken into account and how they were compared. The note to significant differences in Table 1 is also ambiguous since compared groups are not obviously identified and different p values are indicated with respect to the description in Section 4.6.  

 Minor remarks:

- the Title is too long, I suggest to replace it with a shorter one focused on the development of the new mesh and in vivo behavior.

Author Response

In this manuscript, a novel absorbable prosthesis for abdominal hernia is developed, based on electrospun ureidopyrimidinone-polycarbonate (UPy-PC), and compared to a lightweight polypropylene mesh. UPy-PC and PP mesh explants are studied in terms of biomechanical properties. Moreover, muscle atrophy and fat infiltration in host tissue are evaluated. The manuscript presents an interesting idea and valuable results are obtained. However, I have some concerns which the authors should address:

 - Why did you choose Restorelle® mesh as a comparison? Restorelle® is generally used for pelvic organ prolapse repair and not for hernia surgery. The aim of the work seems to be to reduce surgical recurrence and post-surgery pain and discomfort in case of abdominal hernia patients. In this sense, the comparison of UPy-PC with a well-consolidated PP mesh for hernia repair would have been more appropriate. Could you discuss in the Introduction which are the similarities/differences between Restorelle® and other PP hernia meshes? (See, for example: J. Biomed. Mater. Res. Part B: Appl. Biomater., 105B: 892-903, 2017).

Thank you for your comment. Indeed, Restorelle is mostly used for POP surgery, even though it has abdominal wall hernia indications as well. We have chose Restorelle for two reasons. First, because it is ultra-light and has large pores and in theory would have higher biocompatibility. Second, because we aim to develop an implant that is suitable for POP surgery. We have included more data about POP .in the introduction. Now the text reads:

Meshes are used in several conditions, e.g. hernia (where they are the gold standard), and pelvic organ prolapse (POP) (where they are questioned vaginally, but still used abdominally). Non-surgical treatment, such as pelvic floor muscle training for POP shows low-efficacy compared to surgery (1). The high  recurrence rate after native tissue surgical repair remains a challenge (2,3). A significant improvement of hernia and POP repair outcomes were obtained with the use of surgical meshes, however long-term recurrence rates close to 30% remain unacceptably high (4,5).”

On the other hand, the authors are considering the use of absorbable polymers for hernia repair. In this respect, the comparison of degradation behavior in vivo should have been carried out also with a biological mesh. Please, comment on this point and mention the role of biological meshes in the Introduction/discussion too.

Thank you for your suggestions. We have included a paragraph mentioning about the use of biological meshes in hernia repair. It reads on the manuscript:

Use of biological meshes was expected to overcome some of this issues, due to the ability to be remodeled and revascularized, however their inconsistent properties(16) makes difficult to predict their degradation profile (17).”

 - Results: Biomechanical testing. Please, add stress vs. strain plots for PP and UPy-PC explants in the range 0-20% of elongation. Stiffness and Young’s modulus are not enough to compare mesh mechanical behavior.

As suggested, we have included the stress-strain in the range of 0-20% of elongation (Please, see Figure 1).

- Section 4.4 Biomechanical testing. 

1. Why did you test mesh explants in only one direction? Do you expect both PP and UPy-PC to be mechanically isotropic? Did you get any data on PP and UPy-PC meshes before implant?

Yes, we have tested UPy-PC in both directions and it had anisotropic mechanical behaviour. However, it is difficult to define the longitudinal and transverse directions, since the manufacturer did not indicated the direction. PP meshes have isotropic mechanical behaviour. We tested explants in only one direction (muscle working direction). We could not perform the test in both directions due to small sample size. 

 2. Can you explain better how comfort zone stiffness and Young’s modulus were determined? Please, note that I was not able to find any Supplement in my reviewer center. 

            This information is already included in the Material and Methods.

The test was ended when the load fell below 60% of the maximum load. The strain was calculated by dividing the elongation by the cross-sectional area. The Young's modulus in the comfort zone was defined as the minimum modulus noted over an interval of 10-20 % of elongation. Clinically, the comfort zone of the stress-strain curve is considered to be within the physiological range of deformation.”

- Section 4.6 Statistics. This section is too concise and it is not clear which groups were taken into account and how they were compared. The note to significant differences in Table 1 is also ambiguous since compared groups are not obviously identified and different p values are indicated with respect to the description in Section 4.6.  

We have changed it to clarify it. Now the text reads:

All analyses were done with Graphpad Prism 8.0. For FBGC, RAM 11 and CD34 analyses, comparison between groups at different time points were performed using t-test when data was parametric or Maan-Whitney for non-parametric data. Other analyses, which also to native tissue, were analyzed by one-way ANOVA followed by Tukey test. Data are reported as individual values or mean ± SD. Statistical significance level was defined as p<0.05.”< span="">

 Minor remarks:

- the Title is too long, I suggest to replace it with a shorter one focused on the development of the new mesh and in vivo behavior.

As suggested, we have changed the title for “Biomechanical behaviour and biocompatibility of ureidopyrimidinone polycarbonate electrospun and polypropylene meshes in a hernia repair in rabbits”.

Reviewer 2 Report

The manuscript presents an analysis of a novel electrospun material in abdominal wall hernia repair and the comparison with a traditional ultra-lightweight polypropylene mesh, especially in terms of recurrence. 

It is well-written with appropriate language and the methods applied are appropriate for the purpose of the study. The discussion does not over-interpret the results obtained and limitations of the study are mentioned. The manuscript is of interest since the analysis of new materials in abdominal wall hernia repair is needed considering the high recurrence rates still found with the current meshes being used.

However, there are some points that need to be addressed:

1)       Page 2. Lines 46 to 68 require revision to be improved. I include some suggestions in this respect:

 -          Line 47. The authors mention that ´meshes may induce implant-related complications such as pain`. There are many other critical implant-related complications that need to be investigated when testing new materials for abdominal wall repair that should be mentioned. Adhesion formation is a crucial complication since it can lead to bowel obstruction, infertility in gynaecological procedures or chronic abdominal pain. The material rejection due to the host tissue inflammatory response is also an important point. Apart from pain, these implant-related complications should be commented on. There are some recent reviews on this topic that should be included.

 -           Line 50. Reference 4 presents a vaginal mesh implantation. As mentioned in the previous point, there are several recent reviews related to mesh implantation specific for abdominal wall hernia repair that present the factors associated to implant-related complications and should also be mentioned in this point.

 -          Lines 52-58. Please include some references showing evidence for the different statements: ´durable implants induce chronic inflammatory reaction`, ´they do not necessarily induce a functional replacement tissue`, ´the maladaptive remodelling of the less stiff material is characterized by degeneration and atrophy`,…

 -          Lines 65-68. Please include some references related to the use of absorbable polymers and the experimental evidence of the importance of the degradation kinetics in relation to recurrence or host response.

 Some interesting reviews related to these points that should be included are the following:

 -      Todros, S.; Pavan, P.G.; Natali, A.N. Synthetic surgical meshes used in abdominal wall surgery: Part I-materials and structural conformation. J. Biomed. Mater. Res. B Appl. Biomater. 2017, 105, 689–699, doi:10.1002/jbm.b.33586.

-      Gómez-Gil, V.; Pascual, G.; Bellón, J.M. Biomaterial Implants in Abdominal Wall Hernia Repair: A Review on the Importance of the Peritoneal Interface. Processes 2019, 7, 105,  doi: 10.3390/pr7020105

-      Junge, K.; Binnebösel, M.; von Trotha, K.T.; Rosch, R.; Klinge, U.; Neumann, U.P.; Jansen, P.L. Mesh biocompatibility: Effects of cellular inflammation and tissue remodelling. Langenbecks Arch. Surg. 2012, 397, 255–270, doi:10.1007/s00423-011-0780-0.

 2)       Page 2, line 61. There is a typo error in ´however`.

3)       Page 4, line 133. FBGC, please include here that it stands for Foreign body giant cells, since it is the first time it appears in the manuscript.

4)       Page 9, line 269. The study design is not completely clear to me. 12 animals were implanted with UPy-PC and 12 with PP. And samples were harvested at 3 different time points (30, 90 and 180 days, with 3 animals per time point) that would be 9 animals of each type of mesh. Could you please clarify this point?

5)       Page 11, line 370, reference 4. The title of the article is missing.

 Author Response

The manuscript presents an analysis of a novel electrospun material in abdominal wall hernia repair and the comparison with a traditional ultra-lightweight polypropylene mesh, especially in terms of recurrence. 

It is well-written with appropriate language and the methods applied are appropriate for the purpose of the study. The discussion does not over-interpret the results obtained and limitations of the study are mentioned. The manuscript is of interest since the analysis of new materials in abdominal wall hernia repair is needed considering the high recurrence rates still found with the current meshes being used.

However, there are some points that need to be addressed:

 1)       Page 2. Lines 46 to 68 require revision to be improved. I include some suggestions in this respect:

 -Line 47. The authors mention that ´meshes may induce implant-related complications such as pain`. There are many other critical implant-related complications that need to be investigated when testing new materials for abdominal wall repair that should be mentioned. Adhesion formation is a crucial complication since it can lead to bowel obstruction, infertility in gynaecological procedures or chronic abdominal pain. The material rejection due to the host tissue inflammatory response is also an important point. Apart from pain, these implant-related complications should be commented on. There are some recent reviews on this topic that should be included.

Thank you for your suggestions. We have included other implant related complications and added the references as suggested. Now the text reads:

Meshes may induce implant-related complications (IRCs) such as pain, infection, erosion, rejection of the implant and adhesion formation.”

 Line 50. Reference 4 presents a vaginal mesh implantation. As mentioned in the previous point, there are several recent reviews related to mesh implantation specific for abdominal wall hernia repair that present the factors associated to implant-related complications and should also be mentioned in this point. 

 As suggested, We have included other references related to hernia.

 - Lines 52-58. Please include some references showing evidence for the different statements: ´durable implants induce chronic inflammatory reaction`, ´they do not necessarily induce a functional replacement tissue`, ´the maladaptive remodelling of the less stiff material is characterized by degeneration and atrophy`,…

 We have included few references, as suggested.

 -Lines 65-68. Please include some references related to the use of absorbable polymers and the experimental evidence of the importance of the degradation kinetics in relation to recurrence or host response. 

We have included few references, as suggested.

 Some interesting reviews related to these points that should be included are the following:

 -      Todros, S.; Pavan, P.G.; Natali, A.N. Synthetic surgical meshes used in abdominal wall surgery: Part I-materials and structural conformation. J. Biomed. Mater. Res. B Appl. Biomater. 2017, 105, 689–699, doi:10.1002/jbm.b.33586.

-      Gómez-Gil, V.; Pascual, G.; Bellón, J.M. Biomaterial Implants in Abdominal Wall Hernia Repair: A Review on the Importance of the Peritoneal Interface. Processes 2019, 7, 105,  doi: 10.3390/pr7020105

-      Junge, K.; Binnebösel, M.; von Trotha, K.T.; Rosch, R.; Klinge, U.; Neumann, U.P.; Jansen, P.L. Mesh biocompatibility: Effects of cellular inflammation and tissue remodelling. Langenbecks Arch. Surg. 2012, 397, 255–270, doi:10.1007/s00423-011-0780-0.

 2)       Page 2, line 61. There is a typo error in ´however`.

Thank you. We have removed the typo.

3)       Page 4, line 133. FBGC, please include here that it stands for Foreign body giant cells, since it is the first time it appears in the manuscript. 

Thank you. We have included it.

4)       Page 9, line 269. The study design is not completely clear to me. 12 animals were implanted with UPy-PC and 12 with PP. And samples were harvested at 3 different time points (30, 90 and 180 days, with 3 animals per time point) that would be 9 animals of each type of mesh. Could you please clarify this point?

Apologie for this mistake. Indeed, we have used 9 animals per implant. We have correct it on the manuscript. Now it reads:

Eighteen adult New Zealand rabbits (3500g) were randomly divided into two groups, according to the type of mesh (UPy-PC or PP) used.”

 5)       Page 11, line 370, reference 4. The title of the article is missing.

Sorry. We have included this information

 Round  2

Reviewer 1 Report

The authors have responded satisfactorily to all my previous remarks.

Reviewer 2 Report

The comments have been sufficiently addressed and included in this version of the manuscript.